

# Quantitative Comparison of Causal Inference Methods for Climate Tipping Points

Niki Lohmann[1,2,*], David Strahl[3], Annika Högner[4,5], Willem Huiskamp[2], Matthias Boehm[6,7], and Nico Wunderling[1,2,8,*]

[1]Center for Critical Computational Studies (C3S), Goethe University Frankfurt, Frankfurt am Main, Germany
[2]Earth Resilience Science Unit and RD1 Earth System Analysis, Potsdam Institute for Climate Impact Research (PIK), Member of the Leibniz Association, Potsdam, Germany
[3]Institute of Environmental Science and Geography, University of Potsdam, Potsdam, Germany
[4]International Institute for Applied Systems Analysis (IIASA), Laxenburg, Austria
[5]Geography Department, Humboldt-Universität zu Berlin, Berlin, Germany
[6]Technische Universität Berlin, Berlin, Germany
[7]Berlin Institute for the Foundations of Learning and Data (BIFOLD), Berlin, Germany
[8]Senckenberg Research Institute and Natural History Museum, Frankfurt am Main, Germany
[*]Corresponding authors: lohmann@c3s.uni-frankfurt.de, wunderling@c3s.uni-frankfurt.de

**Abstract.** Causal inference methods present a statistical approach to the analysis and reconstruction of dynamic systems as observed in nature or in experiments. Climate tipping points are likely present in several core components of the Earth system, such as the Greenland ice sheet or the Atlantic Meridional Overturning Circulation (AMOC), and are characterized by an abrupt and irreversible degradation under sustained global temperatures above their corresponding thresholds. Causal

inference methods may provide a promising way to study the interactions of climate tipping elements, which are currently highly uncertain due to limitations in model-based approaches. However, the data-driven analysis of climate tipping elements presents several challenges, e.g., with regard to nonlinearity, delayed effects and confoundedness. In this study, we quantify the accuracy of three commonly used multivariate causal inference methods with regard to these challenges and find unique advantages of each method: The Liang–Kleeman Information Flow is preferable in simple settings with limited data availability,

the Peter–Clark Momentary Conditional Independence (PCMCI) provides the most control, e.g., to integrate expert knowledge, and the Granger Causality for State Space Models is advantageous for large datasets and delayed interactions. In general, data sampling intervals should be aligned with the interaction delays, and the inclusion of a confounder (like global temperatures) is crucial to deal with the nonlinear response to (climate) forcing. Based on these findings and given their data masking capabilities, we apply the LKIF and PCMCI methods to reanalysis data to detect tipping point interactions between the AMOC

and Arctic summer sea ice, which imply a bidirectional stabilizing interaction, in agreement with physical mechanisms. Our results therefore contribute robust evidence to the study of interactions of the AMOC and the cryosphere.



## 1 Introduction

The reconstruction of the causal structure of dynamic systems from time series is a challenge in various disciplines like climate science (Runge et al., 2019a), neurosciences (Weichwald and Peters, 2021; Shams and Beierholm, 2022) and economics (Baum-Snow and Ferreira, 2015), where observational data are readily available, but interventional experiments are infeasible or undesirable. Data-driven approaches can reconstruct complex networks, e.g., of brain functions or Earth system phenomena, that could not be derived from expert knowledge. Results of causal analysis can also inform the structure or parameterization of models (Debeire et al., 2025). The application of causal methods to model data may in turn serve as an evaluation of the accuracy of the macroscopic behavior emerging from a model (Nowack et al., 2020). Correlation tests are frequently used to analyze relationships between variables in data, but they do not provide a sufficient theoretical foundation to derive directional causality.

Multivariate causal inference methods provide a theoretical derivation of causality from statistical measures and consider networks of variables and their interactions. Several such methods have been developed in the past decades for different assumptions about the underlying processes (Runge et al., 2019a; Camps-Valls et al., 2023). Previous studies have often tested or compared methods under idealized settings with regard to sample availability (Docquier et al., 2024), interaction strength (Runge et al., 2019b; Liang et al., 2025) and/or network complexity (Nogueira et al., 2022; Assaad et al., 2022).

In recent decades, tipping points have been identified in several core elements of the Earth system, such as the polar ice sheets and the Atlantic Meridional Overturning Circulation (AMOC) (Lenton et al., 2008; Armstrong McKay et al., 2022). These elements are expected to enter tipping processes through destabilizing feedback effects once a respective threshold in forcing has been crossed, directly or indirectly due to global warming. A tipping process is defined by its self-reinforcing dynamics, i.e., even if temperature levels are stabilized above the threshold, the element continuously degrades into a different stable state. This degradation thus becomes irreversible on human time scales once a tipping point is crossed for a significant amount of time. For instance, for the Greenland ice sheet, a tipping process would be characterized by continuous melting due to melt elevation feedback, among other factors (Boers and Rypdal, 2021), which would increase global sea levels by several meters (Morlighem et al., 2017). The AMOC is considered to be forced by freshwater (van Westen et al., 2024; Swingedouw et al., 2022) and warming (Drijfhout et al., 2025; Laridon et al., in review) in its northern convection regions, and could experience a significant weakening or shutdown due to its salt advection feedback (Vanderborght et al., 2025; Caesar et al., 2018; Boers, 2021; Ditlevsen and Ditlevsen, 2023), with severe consequences for the climate of the Northern Hemisphere (Jackson et al., 2015; van Westen et al., 2024).

The resilience of tipping elements has thus become a crucial subject within climate science, communication and the surrounding area of policy. Their relevance is underlined by the dedicated chapter on tipping points in the upcoming IPCC report (AR7, chapter 8: "Abrupt changes, low-likelihood high impact events and critical thresholds, including tipping points, in the Earth system"), large collaborative projects like the Global Tipping Points report (Lenton et al., 2025), and by the recognition of tipping points in the study of economic and social climate change impacts (Trust et al., 2025; Stoerk et al., 2025).





Interactions among tipping elements may further deteriorate the resilience of tipping elements under global warming, several studies have found mostly destabilizing feedbacks (Wunderling et al., 2024), e.g., from meltwater of the Greenland ice sheet to the AMOC (Swingedouw et al., 2022; Klose et al., 2024). Research on these tipping point interactions is still in an early stage and even lacking entirely for several hypothesized tipping element interactions. However, some initial work has been conducted using Earth system models of intermediate complexity (EMIC) (Willeit et al., 2023; Kaufhold et al., 2025; Sinet et al., 2025),

as fully complex Earth system models are only starting to represent elements like ice sheets and vegetation dynamically.

Several causal methods are suitable to detect interactions between variables in dynamic systems. Applying these methods to time series data of (potentially) interacting tipping elements may provide a promising data-driven approach to estimate tipping point interactions without the usage of climate models. Causal analysis has so far only been applied to climate tipping points directly in a study where the PCMCI method was used to detect a stabilizing interaction from the AMOC to the Southern

Amazon rainforest (Högner et al., 2025). However, so far there is no systematic assessment of different causal methods applied to synthetic data that displays nonlinear behavior resembling the dynamics of tipping elements and that can be designed to exhibit characteristics typical for Earth observation data.

With this study, we aim to estimate the reliability and robustness of three causal methods in the context of climate tipping points, which pose some specific challenges:

– As the period of modern observations is short compared to the timescales of tipping elements, the number of available samples is low, often below 1000 (Högner et al., 2025; Kretschmer et al., 2016)

– Interactions between analyzed elements may be weak or highly delayed due to the atmospheric or oceanic transport required to propagate interaction effects (Di Capua et al., 2023)

– Large and dense networks of variables may be of special interest, e.g., due to regional tipping patterns (Runge et al.,
70    2015)

– Global warming influences all climate tipping elements, bringing them closer to their tipping thresholds. This prevents stationarity in the dynamic systems and contributes some common noise to all time series, both of which may complicate causal analysis.

We therefore conduct experiments to explore the role of these parameters in three causal inference methods, selected for their
wide use in the literature and their capacity for multivariate analysis in limited datasets (e.g., in contrast to neural network methods, which require much larger datasets): The Liang–Kleeman Information Flow (LKIF) (Liang, 2021), the Peter–Clark Momentary Conditional Independence (PCMCI) (Runge et al., 2019b), and the State Space Granger Causality (GCSS) (Barnett and Seth, 2015).

We generate data from a network model of differential equations and rate the detection capabilities of each method quan-
titatively by comparing the statistically significant detected links with the true constellation of interactions in the underlying model.





We derive three core recommendations from our results, which should provide some guidance on the application of causal methods to climate tipping points, especially with regard to the required assumptions on data and the underlying physical system.

We further conduct an applied experiment to demonstrate the usage of the analyzed causal methods on a tipping point interaction. With freshwater influx considered an important factor for AMOC stability (Weijer et al., 2019) and given previous ambiguous results on the role of Arctic sea ice (Weijer et al., 2022), we decide to focus on interactions between the AMOC and Arctic summer sea ice (ASSI). After conducting appropriate preprocessing steps of available time series data, and with the inclusion of Arctic temperatures as a potential confounder, we find robust bidirectional stabilizing interactions between the

elements with the PCMCI method. These results imply that a weakening of the AMOC would slightly increase summer sea ice concentration, and a degradation of summer sea ice would strengthen the AMOC. Both of these effects can be explained by physical mechanisms, as an AMOC weakening would lead to reduced heat transport into the Arctic and subpolar seas, and reduced sea ice cover may lead to increased surface cooling, and therefore increased convection, in the subpolar North Atlantic or Nordic seas.

This paper is structured as follows: Section 2 describes the methodology of our experiments, the utilized causal methods and the data generation process. In Sect. 3, we present the results of our comparative experiments on synthetic data under various conditions expected to be present in physical tipping elements. In Sect. 4, we discuss the results and derive recommendations for the application of causal methods in the study of climate tipping points and their interactions. An applied experiment is presented in Section 5, where we apply our recommendations to a hypothesized interaction between the AMOC and Arctic sea

ice. Section 6 provides conclusions and outlooks for further work.

## 2 Methods and Data

For our model experiments, data are generated from a network model, as visualized in Fig. 1, where each node corresponds to a nonlinear variable, and each edge to an interaction. For one of the experiments, the network is extended to contain a confounder interacting with all variables, i.e., a variable that is not known to the causal method introduces common noise in

several variables, potentially leading to a wrong attribution of the common noise as a causal effect between known variables. The generated time series data are fed into three causal algorithms, each of which determines a graph of significant interactions. Lastly, the predicted graph is compared to the true network model used for data generation as an evaluation of detection accuracy.

In the following, we first introduce the data generation model, then describe each of the utilized causal methods and their

underlying concepts of causality. Lastly, we introduce the metrics of detection accuracy used in the experiments.





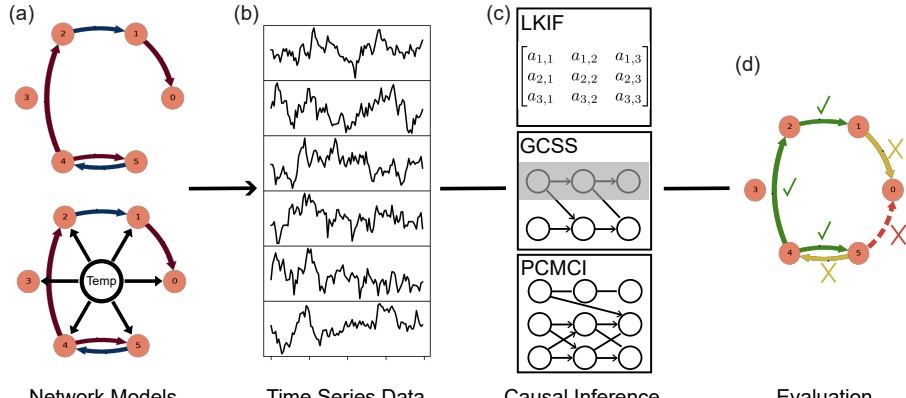

**Figure 1.** Conceptual overview of the methodological approach. (a) A network of interacting nonlinear dynamic systems generates (b) time series data for each variable. (c) Three causal methods are applied to the data to detect interactions, (d) their results are then checked against the ground truth network for accuracy evaluation. The chosen visualizations of methods in (c) are abstractions of their underlying modeling approach. In (d), green arrows indicate true positives, yellow arrows indicate false negatives, the red arrow indicates a false positive.

## 2.1 Data

Synthetic data are generated from a set of cubic stochastic differential equations (SDEs) each of which can exhibit a fold bifurcation, i.e., a tipping process once a certain threshold is crossed.

$$\dot{x}_i = -x_i^3 + x_i + c + \sum_{j \neq i} s_{j,i}(x_j - 1) + \sigma dW. \tag{1}$$

The term $\sum_{j \neq i} s_{j,i}(x_j - 1)$ adds linear interactions between variables, we only impose nonlinearity on the dynamics of the variable itself through the cubic term $-x_i^3$. The Wiener noise $dW$ is generated separately for each variable and scaled by $\sigma$. The variable $c$ introduces external forcing, in models of climate tipping elements this may resemble climate forcing.

In our experiments we also analyze the impact of delays on the detection of causal interactions. For some fixed delay $\tau$, the delayed differential equation (DDE) for the model then becomes:

$$\dot{x}_i(t) = -x_i(t)^3 + x_i(t) + c + \sum_{j \neq i} s_{j,i}(x_j(t - \tau) - 1) + \sigma dW. \tag{2}$$

Starting at $c = 0$, each variable is initialized to the stable equilibrium at $x = 1$, with the other equilibrium at $x = -1$. Once the sum of interactions and $c$ crosses the threshold of $-\sqrt{\frac{4}{27}}$, only the negative stable equilibrium remains and the tipping process into this state starts (Klose et al., 2020). The cubic differential equation is one of the least complex systems showing this tipping behavior and therefore used for abstract modeling of tipping points in the climate system, among other areas
(Wunderling et al., 2021; Bdolach et al., 2025).

The interactions between cubic models are conceptualized as the edges in a graph network model. We manually designed networks of different sizes and densities, where a dense system has a higher connectivity of nodes, with a focus on feedback





loops. We also chose the signs of interactions (i.e., negative or positive interactions) such that the resulting systems are stable under all parameterizations used in the following experiments. The manually designed systems can be found in Fig. A1, from
which we derived systems of other sizes, see Appendix A.

## 2.2 Causal Inference Methods

For our analysis, the main task of the utilized methods is the identification of significant causal links from time series, this task is also referred to as causal discovery. All of the following methods provide statistical significance tests at some confidence level $\alpha$, indicating the error rate. Although the estimation of causal effect strengths is highly relevant to the interpretation
of results, the different concepts of causality employed by these methods do not allow for a quantitative comparison of the accuracy of the strength estimations. All methods can handle multivariate systems, either through fitting of vectorized models or through the inclusion of confounder variables in otherwise bivariate statistical tests. Every method also requires a maximal time delay at which interactions should be detected. We selected the following methods as they fulfill formal requirements (e.g., the possibility of significance tests), their performance converges on dataset sizes we deem realistic for real-world applications
and they are established in literature. In Appendix B, we provide a mathematical derivation and explanation of how these methods can be fitted to the data generated from the nonlinear dynamic systems of Eq. 1.

### 2.2.1 Peter–Clark Momentary Conditional Independence (PCMCI)

The PCMCI algorithm was introduced by Runge et al. (Runge et al., 2019b) and combines the Peter–Clark (PC) algorithm with a momentary conditional independence (MCI) test. It has been successfully applied to various areas of climate science,
including tipping points (Högner et al., 2025), atmospheric (Di Capua et al., 2024), oceanic (Falkena et al., 2025), cryosphere (Kromer and Trusel, 2023) and atmosphere–ocean interactions (Docquier et al., 2024).

In the PC phase of the algorithm, conditional independence tests are applied iteratively between a variable and past time steps of other variables, conditioned on an iteratively growing set of the most significant causal parents of the target variable.

Once each variable has reached a stable set of causal parents, the MCI phase conducts final tests conditioned on the causal
parents of both variables involved in a hypothesized causal link.

Although the conditional independence tests appear to be pairwise in the strict sense, they provide multivariate analysis through the iterative changes to conditional variables: The method starts from a fully connected graph of interactions (between all variables, each at all analyzed time shifts), and then prunes false positives in the PC phase. The network is therefore optimized only by the iterative selection of conditional variables. On the other hand, the pairwise tests allow for a granular
treatment of variables:

- Known causal connections can be imposed, or interactions can be prohibited if they are considered unrealistic or impossible.

- Sections of the data can be left out (masking), e.g., for missing samples or restriction to certain time periods.

- Masking can be applied exclusively when a variable is tested as a causal parent (or causal child, or conditional variable).





In this work, we utilize the PCMCI+ version of the algorithm, an extension that includes contemporaneous links (Runge, 2020). The underlying conditional independence test can also be chosen according to domain knowledge. In this study, we analyze the linear partial correlation test as it is the least computationally intensive, but nonlinear or model-agnostic tests are also available for usage with PCMCI (Runge, 2020), a comparison to other independence tests can be found in D. The significance level is applied to every independence test throughout the iterative phase and only provides an orientation about

the significance of the final result, rather than an exact estimate of the error rate.

### 2.2.2 Liang–Kleeman Information Flow (LKIF)

The Liang–Kleeman Information Flow (LKIF) has only been applied to climate research very recently, with application to the cryosphere (Docquier et al., 2022) and land–atmosphere interactions (Zhou et al., 2024; Shao et al., 2024). A comparative study by Docquier et al. (2024) found similar reliability of PCMCI and LKIF under idealized settings.

The LKIF method takes an information-theoretic approach to causality, by which the contribution of entropy from one variable to another constitutes its causal effect (Liang, 2021).

In practice, a low complexity model is fitted to the time series data. From this model, the entropy contribution is derived in the style of an intervention experiment, i.e., it is measured how the entropy of variables would change if one variable was missing from the model.

The LKIF method is implemented with a linear SDE model, from which entropy transfers can be derived analytically (Liang, 2021).

$$\dot{X} = AX + b + dW. \tag{3}$$

Time lag analysis is implemented by shifting any single input time series by a given number of time steps. For a causal link to be considered as detected in our evaluation it is sufficient to be statistically significant at any time lag.

### 2.2.3 Granger Causality for State Space Models (GCSS)

The Granger Causality for State Space Models (GCSS) method has so far seen little recognition in climate research, instead it has been widely used in the analysis of neural dynamics (Binns et al., 2025; Yue et al., 2025; Barnett and Seth, 2016).

Granger Causality (GC) states that a variable is causal to another if its existence helps to predict the affected variable. Like for LKIF, the measurement of GC can be thought of as an intervention experiment, where one variable is removed and other

variables are predicted less accurately because of it.

Barnett and Seth apply GC to a state space model (Barnett and Seth, 2015), which consists of an observation and a hidden process. It is time-discrete and able to describe vector-autoregressive processes with moving average components, i.e., a direct effect of past noise on current time steps.

$$
\begin{aligned}
x_{t+1} &= Ax_t + u_t && \text{Hidden State Transition} \\
y_t &= Cx_t + v_t && \text{Observation}
\end{aligned}
\tag{4}
$$





While the chosen GCSS implementation does not provide information on the time distribution of causal effects, its hidden state dimension allows for an implicit time shifted analysis: The model can keep track of older values of some variable $x_{i,t}$ by fitting a variable to behave as $x_{j,t+1} = x_{i,t}$, thereby giving access to the value of $x_{i,t}$ at one time step later and so on. If the maximal time lag of interactions is known, the maximal hidden state dimension can be determined accordingly.

We implement a $\chi^2$-test for significance as suggested by the authors (Barnett and Seth, 2016, 2019), and scale its degrees of
freedom with the dimension of input data (i.e., variable count).

## 2.3  Metrics

For easy comparison, we use a single metric that contains information on true and false positives (TP/FP) and true and false negatives (TN/FN). The Matthews Correlation Coefficient (MCC) is considered a good candidate for such a combined metric in comparison to other frequently used metrics in binary classification (Chicco and Jurman, 2020). The inclusion of all four
basic metrics avoids biases towards denser or less dense systems: In a network with a sparser graph matrix, the absolute number of false positives would likely increase (assuming a constant false positive rate), but other popular metrics like the F1 score do not include true negatives, while the MCC rewards true positives and true negatives symmetrically. The MCC is defined using the abbreviations for the basic metrics above:

$$MCC = \frac{TP \cdot TN - FP \cdot FN}{\sqrt{(TP+FP) \cdot (TP+FN) \cdot (TN+FP) \cdot (TN+FN)}}. \tag{5}$$

For more detailed analysis, we also provide true and false positive rates, which are calculated as follows:

$$TPR = \frac{TP}{TP+FN}$$
$$FPR = \frac{FP}{FP+TN}. \tag{6}$$

## 3  Results

For the following experiments, we use a default configuration of data and method parameters that we deem realistic for a setting in the analysis of climate tipping elements, and physical systems more broadly. Where a parameter is not varied explicitly, it is
fixed to its default value listed in Table C1. For each parameter varied in an experiment, its range is given as well. The default model network contains six variables with a low density of interactions. One variable does not interact at all, the remaining variables are connected through five causal links. The network is visualized in Fig. A1c. Every experiment is run 100 times and we determine our metrics for each run, then show the average and standard deviation of those scores.

Supplementary results on different conditional independence tests for PCMCI are provided in Appendix D, a comparison of
runtimes of the causal methods is found in Appendix E.





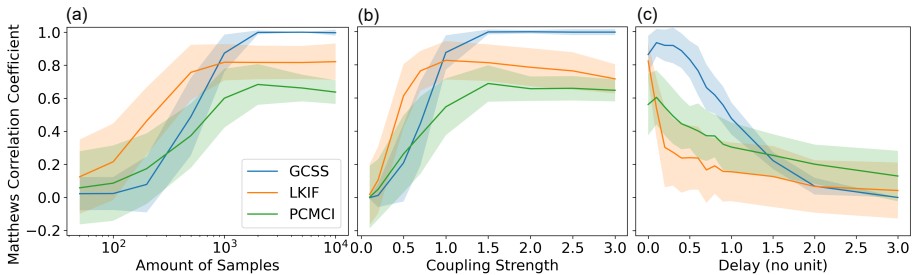

**Figure 2.** Relationship between basic data parameters and prediction scores of the algorithms. (a) Prediction scores increase between 200 and 1000 provided samples, with LKIF being most reliable in intermediate numbers of samples, and GCSS most reliable for larger numbers. (b) Similar development for coupling strengths between variables, where scores increase massively from strengths of 0.1 and 1.0, with a higher convergence for GCSS than for LKIF and PCMCI. (c) Delays of interactions decrease performance for all algorithms, GCSS can handle small delays well. Shaded areas indicate one standard deviation.

## 3.1 Results for Data Parameters

Figure 2a shows a sharp rise in accuracy between 100 and 1000 provided samples for all methods, with LKIF being more reliable than the other two methods in this order of magnitude of available samples. For larger numbers of samples, the GCSS method yields nearly perfect prediction scores, while LKIF and PCMCI remain at MCCs around 0.8 and 0.6, respectively.

The scores for varying coupling strengths between the variables develop in a similar way (Figure 2b): At low coupling strengths between 0.5 and 1, the LKIF method shows the best performance of the three methods, for higher coupling strengths the GCSS method predicts interactions perfectly again, both PCMCI and LKIF remain with imperfect scores for any coupling strength. The abstract size of coupling strengths can be illustrated with its impact on signals: The sampling rate of 10 samples per time unit means that the time-discrete approximation of the system (Equation B2) results in a linear interaction coefficient that is ten times smaller, i.e., for a coupling strength of 0.5, the linear interaction coefficient is 0.05. Given the identical noise

scale of all variables, a coupling strength of 0.5 would therefore imply a signal-to-noise ratio of $\frac{1}{20}$ for the detection of this causal interaction.

The results in Fig. 2a-b demonstrate the tradeoff made by the GCSS method, which comes with a moderately higher data intensity (and lower tolerance for ambiguity, w.r.t. coupling strength), but achieves nearly perfect results under ideal circum-

stances. The strict model assumptions for LKIF can explain its early convergence at high but imperfect accuracy, i.e., very few parameters need to be fitted (which leads to good results with few samples), but they lack explanatory power to capture the data entirely, no matter how many samples are provided.

When a delay is applied to all interactions (see Equation 2), performance drops rapidly for the LKIF algorithm even at a delay of 0.1 time units, i.e., one sample, see Fig. 2c. The PCMCI algorithm experiences a gradual decrease in prediction

accuracy. The GCSS method can handle small delays very well, and only becomes less reliable from a delay of five samples onwards.





These qualitative differences can be explained by the underlying assumptions: The LKIF model derives time-lagged causality from a shift of input time series, which can only account for unidirectional time shifts, but not delayed feedback loops. The time-discrete methods of PCMCI and GCSS perform better for time lags, we consider it likely that GCSS provides higher explanatory power due to its state space model, which also comes at higher data intensity to achieve the scores seen in Fig. 2c.

### 3.2 Results for Network Configurations

Figure 3 shows results for the different networks of variables separately for each variable. The corresponding manually designed systems can be found in Fig. A1 and vary in the number of variables and the density of edges in the graph, i.e., the number of interactions relative to the number of nodes. The GCSS method shows a large decrease in performance for larger networks, irrespective of the density. GCSS consistently shows slightly higher detection capabilities for networks of low density (see Fig. 3a). LKIF and PCMCI only show a slight decreasing trend in prediction scores for higher numbers of variables. LKIF detects networks of high density more accurately (see Fig. 3b), while PCMCI is ambiguous in this regard, as it scores better on networks of low density than of high density for small numbers of variables, but high density networks seem to converge at slightly higher scores for higher numbers of variables (see Fig. 3c).

In these and other tests, we observed that the optimal choice of a significance level $\alpha$ for any method depends heavily on the given network size and density as well as on other data parameter choices. Therefore, using a different $\alpha$ than in the experiment may yield a qualitatively different accuracy landscape. However, our choice of $\alpha = 0.05$ is based on its common choice in the literature to ensure reasonable confidence.

### 3.3 Results on External Forcing

In the final experiment, we apply an external forcing to all variables, driving them closer to a tipping point. The forcing parameter $c$ in Equation 1 is now time-dependent (with $t \geq 0$), rising linearly for 50 units of time, and staying constant for the remaining 50 units of time at a maximum value of $c_{\lim} \cdot fs$ (forcing strength, given on the x-axis of Fig. 4), which is derived from the bifurcation point of the equation (without interactions).

$$c(t) = \begin{cases} \frac{c_{\lim}}{50} \cdot t & \text{if } t < 50 \\ c_{\lim} & \text{else} \end{cases}, \qquad c_{\lim} = -\sqrt{\frac{4}{27}} \cdot fs \tag{7}$$

The forcing parameter $c$ could resemble the impact of global warming on climate tipping elements. This variable is also referred to as a confounder, as it influences all time series without necessarily being part of the causal analysis. We then compare the detection scores between the exclusion and inclusion of the confounder in the causal analysis.

When the forcing level does not cross any tipping points, the false positive rate still rises by several standard deviations for the LKIF algorithm in comparison to the absence of forcing. However, when the forcing variable is included in the causal analysis, the forcing strength does not have an influence on the false positive rate of LKIF. Similarly, the true positive rate of GCSS drops for an unknown confounder, but remains unchanged if the confounder is included in the causal analysis.

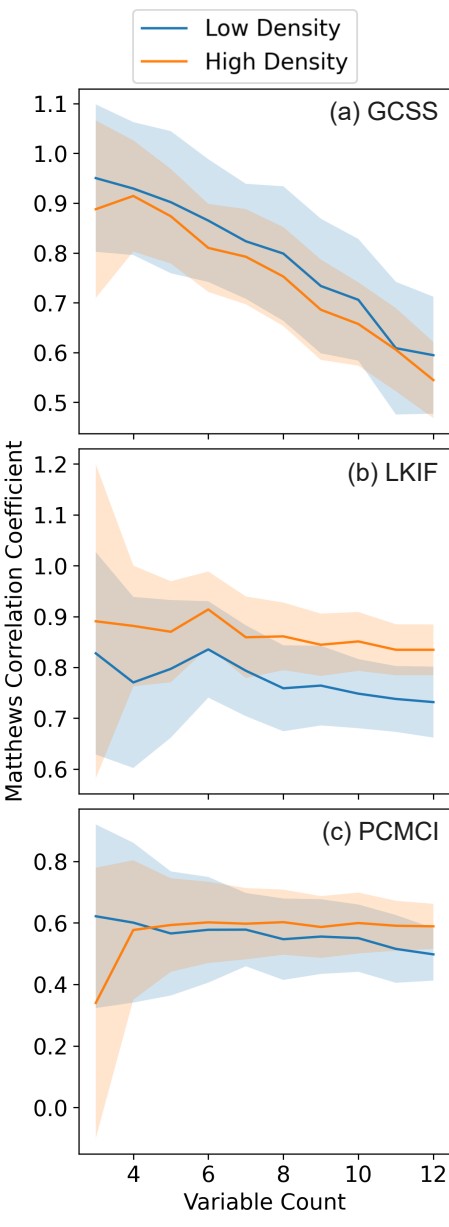

**Figure 3.** Prediction capabilities of the (a) GCSS, (b) LKIF, and (c) PCMCI methods for varying number of variables and interactions in the underlying model system. The prediction score of the GCSS method drops nearly linearly with increased variable count, while LKIF and PCMCI only see minor decreases. Density appears to play a smaller role, with GCSS performing better for low density networks and LKIF better for high density networks.

When a tipping point is crossed (forcing strength $\geq 0.6$, right hand side in Fig. 4), one or more variables undergo abrupt tipping processes into their alternative stable states. Due to the interactions between tipping elements in this model, the forcing




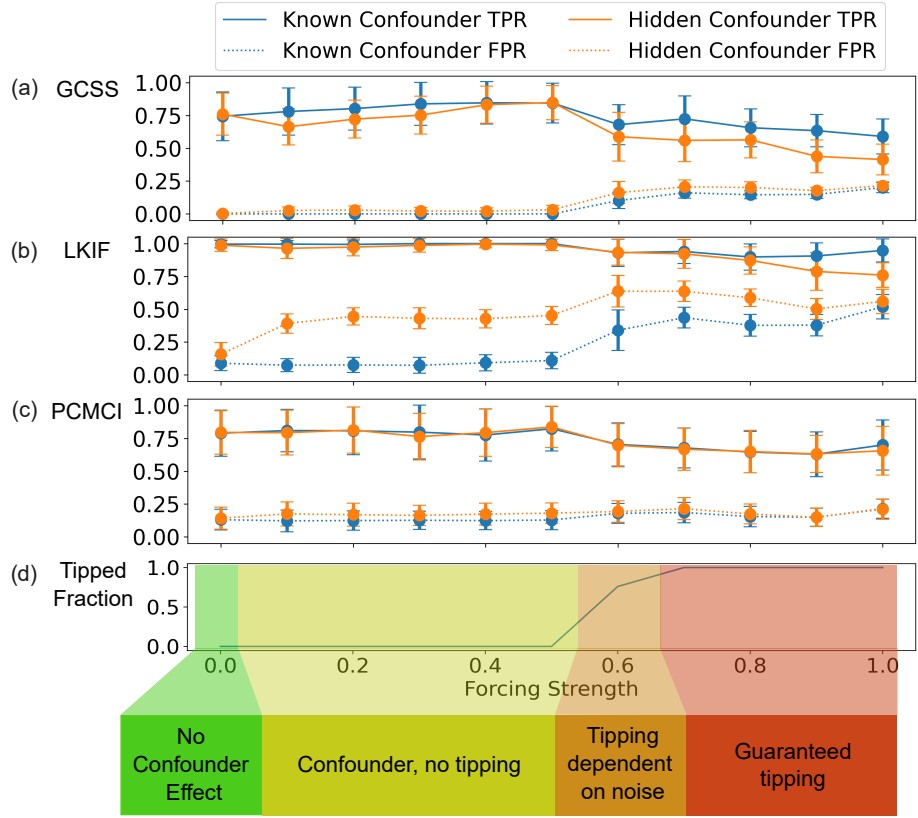

**Figure 4.** The influence of confounders on the detection of tipping point interactions. (a-c) Prediction capabilities under increasing confounder strength, with true and false positive rates for causal analysis including and excluding the confounder variable, respectively. Vertical lines indicate one standard deviation. For the GCSS method, the inclusion of a confounder mainly increases the true positive rate. The LKIF method sees a large decline in false positives, the PCMCI method does not show a clear effect. (d) Fraction of time series that show tipping processes, as noise plays a larger role around the threshold. Shading indicates the color-coded confounding scenario.

strength required to tip at least one element is lower than for an isolated element (which would tip at a forcing level of 1). In Equation 1, the previous equilibrium around $x = 1$ ceases to exist as the function values become strictly negative for any $x > 0$ due to the strong effect of $c(t)$.

GCSS and LKIF show both a drop in true positives and a rise in false positives. While the inclusion of the confounder remains beneficial, a false positive rate $> 25\%$ for LKIF sheds doubt on any results under such a setting. Especially in applications to physical systems, which are expected to have relatively sparse causal interactions, a false positive rate of $25\%$ may produce more false positives than true positives in absolute terms. For GCSS, the low rate of true positives also would make it difficult to draw robust conclusions from an experiment under such conditions.




Notably, the PCMCI method does not show any clear trends from the inclusion of a forcing variable. Only in systems with tipping dynamics, the true positive rate drops by about one standard deviation. While PCMCI is the least affected by confounding, its overall performance implies it remains the least reliable choice in any of the scenarios.

## 4   Discussion

In general, our results on the relationship of sample count and detection capabilities are encouraging for the usage of these methods, even under settings with limited data availability, e.g., around 500 samples. Similarly, one can expect to reliably detect causal relationships with a coupling strength of 0.7 or more using the LKIF method. Notably, the GCSS method is a very reliable choice for settings with either large sample counts or strong causal interactions, which could make it more interesting for e.g., analysis of time series from long runs of climate models.

The results on interaction delays call more attention to the choice of a causal method in the application to physical systems. Every physical interaction of tipping elements operates at some time delay, simply in order to transfer information through various means (e.g., transport of heat, pressure, precipitation, salinity, etc.). Given our presented results on the influence of time delays on detection capabilities, we give the following recommendation:

**Recommendation 1:** The sampling rate of observations should match the scale of time delays of interactions.

A possible explanation for the large performance drop of LKIF under delays is its strict assumption of an underlying SDE, i.e., a time-delayed effect (representable by a DDE) cannot be reproduced inside the model. Even though the input time series are shifted in time for time-lag analysis, such shifts cannot reproduce delays in feedback loops. The good detection capability of GCSS is likely caused by its larger model complexity. It explicitly allows for the representation of delayed variables, and computes the significance of a variable's causal effects across all time lags.

The assumption of stationarity is crucial to the linearization of the chosen dynamical system, as described in Section B. In the analysis of climate tipping points, one has to assume a violation of the stationarity assumption given that the expected levels of warming in the next decades could be sufficient to cross multiple tipping points (Armstrong McKay et al., 2022).

**Recommendation 2:** In the presence of a destabilizing forcing variable, the inclusion of this forcing variable in causal analysis is strongly recommended. Even for a nonlinear system response to forcing, this inclusion mitigates any negative effects of forcing on prediction capabilities, as long as the analyzed dynamic systems do not enter a tipping process.

We do not recommend any of the discussed causal methods if a tipping process is present in analyzed data, assuming similar other parameters in terms of network size, sample count and coupling strength.

Throughout the experiments, the different methods showed respective strengths and weaknesses. We summarize them here in order to give advice on the choice of a causal method in a concise manner.

**Recommendation 3:** The choice of a causal method fitting to the known conditions is crucial for successful causal analysis. Two methods have concrete niches for their application: The GCSS method is advisable to be used if one can assume significantly delayed causal effects, and for settings with large sample counts, few variables and/or strong interactions. The PCMCI method offers the largest flexibility and should be used when domain knowledge needs to be integrated into causal analysis,





e.g., prohibiting impossible links or masking data. The LKIF method offers the best performance in most cases that do not
fall under any of the above constraints. Note that the usage of multiple of the presented methods can serve as a robustness test
under conditions that do not fall clearly into any category.

Previous assessments of PCMCI (with the partial correlation test) confirmed its weaknesses in nonlinear settings (Delforge
et al., 2022; Liang et al., 2025). Our results also highlight a significant difference between the performance of PCMCI and
315 LKIF in most parameter settings, which extends a previous comparison study by Docquier et al. (2024), which found a more
similar performance of the two methods in linear systems and for larger numbers of samples. Our results confirm the difficulties
with time lags for the LKIF method identified in the former study. The existing literature provides ambiguous results on the
connection between the number of samples and prediction performance, with some studies showing a relatively small increase
in prediction scores for growing sample size (Assaad et al., 2022), others show trends that are similar to our results, but often
with better scores at low numbers of samples (Runge, 2020; Liang et al., 2025). We consider it likely that this difference is due
to the choice of linear, time-discrete models and larger interaction coefficients in these studies.

## 5 Application to Climate Tipping Points

We demonstrate the application of causal inference methods to climate tipping points on the interactions between the AMOC
and Arctic summer sea ice. Following our third recommendation, we apply PCMCI and LKIF, as both can restrict analysis to
325 seasonal sections of time series, which is not possible with GCSS in a straightforward way. Even though Arctic summer sea
ice is not considered a tipping element of the climate system (Lenton et al., 2025), abrupt shifts in Arctic summer sea ice have
been detected in recent studies on CMIP6 models (Terpstra et al., 2025; Angevaare and Drijfhout, 2025), indicating at least a
nonlinear relationship between forcing levels and sea ice coverage. Arctic summer sea ice extent has decreased drastically due
to global warming over the past decades, and an ice-free Arctic in summer is projected to occur by 2050 (Jahn et al., 2024).
The current literature is mostly based on model experiments and suggests a stabilizing effect of a weakening AMOC on
Arctic sea ice on annual to decadal timescales due to the reduced northward heat transport of the AMOC (Mahajan et al., 2011;
Docquier and Koenigk, 2021; Weijer et al., 2022). The effect of Arctic sea ice decline on the AMOC is subject to multiple,
partly competing explanations: The increasing amount of freshwater from sea ice melt is considered to have a destabilizing
effect on the AMOC (Li et al., 2021; Liu and Fedorov, 2022; van Westen et al., 2024). On the other hand, the larger area of
335 exposed sea surface may lead to an increase in heat loss (Wu et al., 2021) and therefore a stronger convection, or an increase
in radiative warming (Sévellec et al., 2017; Jenkins and Dai, 2022), which would weaken convection.

For our data–driven analysis, we use a sea surface temperature (SST) fingerprint of the AMOC established by Caesar et al.
(2018) based on ERA5 data (Hersbach et al., 2023) as previously used in data-driven causal analysis by Högner et al. (2025),
and reanalysis data of Arctic sea ice concentration (E.U. Copernicus Marine Service Information (CMEMS). Marine Data
Store (MDS), 2024b) based on the neXtSIM model (Williams et al., 2021). We aggregate the Arctic sea ice concentration
over a 180° slice (90° W to 90° E), i.e., on the side facing the Atlantic, and we only include cells above the 66th percentile



of variation of sea ice concentration, as we assume for them to show the strongest interaction signal (e.g., the meltwater flux would be strongest from these cells).

Arctic sea ice variability is largely driven by atmospheric temperatures in the Arctic (Olonscheck et al., 2019; Chen et al., preprint) and oceanic heat transport (Docquier et al., 2022). In turn, Arctic sea ice concentration plays a major role in the regional climate of the Arctic (Carvalho and Wang, 2020), a reduction in Arctic summer sea ice was found to explain temperature increases especially in autumn and winter (Huo et al., 2025). To rule out a potential confounding effect of temperature conditions in the Arctic on both Arctic sea ice and the AMOC, we use an aggregate of sea surface temperature and ice surface temperature data of the Arctic Ocean (E.U. Copernicus Marine Service Information (CMEMS). Marine Data Store (MDS), 2024a). All variables are detrended and we remove seasonal trends, i.e., for each sample point, the average value of its month is subtracted.

As mentioned, the most drastic effects of global warming on Arctic sea ice are observed in summer. Additionally, the described physical mechanism of meltwater influx from Arctic sea ice to the AMOC also warrants a focus on the summer season of Arctic sea ice. We therefore restrict analysis to the period from March to September, i.e., the months from the maximum to the minimum of Arctic sea ice extent (Stroeve and Notz, 2018).

This is possible in PCMCI and LKIF by masking data, i.e., certain time periods of data are ignored in causal analysis. For the PCMCI method, we can further define these masks to be applied only to the causal parent, i.e., we search for causal effects originating in the summer months. Therefore our analysis may only detect interactions from the Arctic summer sea ice to the AMOC, but may include effects of the AMOC on Arctic sea ice changes later in the year. The LKIF method simply drops all masked samples.

Given a relatively short observation period of 31 years, it is infeasible to conduct analysis on annual data. In order to still capture the inherent time scales of oceanic and cryospheric processes (following recommendation 1), we decided for a monthly timescale. Our analysis therefore deals with a sample count of 217 samples for LKIF, while PCMCI may involve delays of up to 5 months (beyond the mask on the causal parent), resulting in the full 372 valid samples (in causal children and conditional variables). Following our results in Fig. 2a, we are sufficiently confident to draw consistent conclusions with additional robustness tests (see Fig. F1b-e), although a higher sample count would reduce uncertainty.

With the PCMCI method, our causal analysis reveals a bidirectional interaction between Arctic summer sea ice and the AMOC, see Fig. 5. The effect of the AMOC on Arctic sea ice is found to be stabilizing, i.e., a weakening of the AMOC would increase Arctic sea ice concentration. Arctic summer sea ice is found to have two effects on the AMOC at different time delays, the stronger one is a stabilizing effect on the AMOC with a delay of one month, i.e., a loss of Arctic summer sea ice would strengthen the AMOC after one month. Additionally, PCMCI shows a weaker and more delayed destabilizing link (time lag of five months, see arrow labels in Fig. 5).

These links are considered stabilizing in the sense that the degradation of one element increases the tipping resilience of the other, affected element. However, the full circular interaction still implies a destabilizing loop (i.e., a weaker AMOC leads to more Arctic sea ice, which leads to an even weaker AMOC).




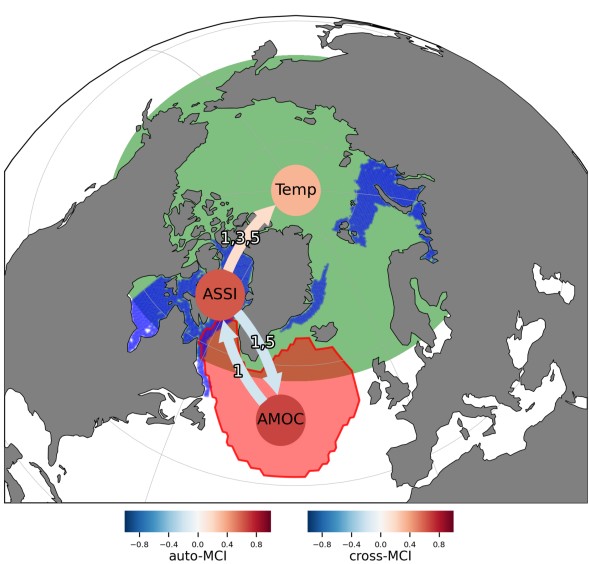

**Figure 5.** Causal network between Arctic summer sea ice (blue), the AMOC (red), and Arctic temperatures (green), as detected by the PCMCI method. A bidirectional stabilizing interaction of Arctic sea ice and the AMOC is found, while a confounding effect of Arctic temperatures is refuted. Note that the positive interaction coefficient from Arctic summer sea ice to the Arctic temperatures implied by the arrow shading does not correspond to the strongest effect of the three delays, for which the coefficient is negative, implying an inversely proportional effect.

The LKIF method (Figure F1a) also shows a link from the AMOC to Arctic summer sea ice, implying an information transfer of 9.5%. It does not detect any further causal links.

However, we have several reasons to consider PCMCI the more reliable approach in this application example:

– LKIF considers a significantly lower amount of samples in its analysis.

– There are inherent delays in the underlying physical mechanisms, which may make PCMCI more reliable, see Fig. 2c.

– Such delays are further supported by the findings of PCMCI, with delays up to 5 months.

We therefore focus our further analysis on the results of the PCMCI method.

With PCMCI, we find a causal effect from Arctic summer sea ice to the Arctic temperatures. Effects are found at delays of one, three and five months, of which the delay at three months is the strongest by a factor of five. This effect confirms the
literature findings by which a decline in Arctic sea ice increases Arctic temperatures. Although the causal effect of temperatures on Arctic sea ice as in Docquier et al. (2022) is not detected in this experiment, the integration of Arctic surface temperatures underlines that our detected effects occur between Arctic sea ice concentration and the AMOC rather than between sea and ice surface temperatures in the Arctic (as measured by the Arctic temperature variable) and the North Atlantic (as measured by the AMOC fingerprint).





With a causal strength estimation method (linear mediation, see Runge et al. (2015)), we can estimate that the stabilizing effect from the AMOC to Arctic sea ice would result in an increase of 0.1 percent points in Arctic sea ice concentration for every 1 Sverdrup (Sv) of AMOC weakening. Given that model and observation estimates of the AMOC strength vary between 15 and 25 Sv (van Westen et al., 2025; Drijfhout et al., 2025; Frajka-Williams et al., 2019), this effect would be very weak even in the case of a shutdown of the AMOC.

In the other direction, the strength estimation for the stabilizing link from Arctic summer sea ice to the AMOC implies that for every ten percent points of Arctic summer sea ice concentration loss, the AMOC would be strengthened by 0.61 Sv one month later. The destabilizing link at five months delay would imply a weakening of the AMOC by 0.14 Sv for the same loss of Arctic summer sea ice. The contradictory direction of these links may be explained by the competing physical effects on different timescales: The freshwater influx destabilizing the AMOC may take several months due to the required oceanic transport, while increased heat loss of surface waters to the atmosphere is a more immediate effect of reduced sea ice concentration and may explain the stabilizing link found here.

In this applied example of two of the three tested algorithms, a data-driven analysis of interactions between Arctic summer sea ice and the AMOC on time scales of months, we find indications for all three interaction effects suggested by the literature:

–   A weakening of the AMOC would stabilize Arctic sea ice due to a reduced northward heat transport (Mahajan et al., 2011; Docquier and Koenigk, 2021; Weijer et al., 2022). This effect is confirmed by PCMCI and LKIF.

–   Loss of Arctic summer sea ice may stabilize the AMOC, as increasingly exposed ocean surfaces may lose more heat to the atmosphere (Wu et al., 2021).

–   The freshwater influx from melting Arctic summer sea ice may destabilize the AMOC, although likely on larger timescales (Li et al., 2021; Liu and Fedorov, 2022; van Westen et al., 2024). Model results are therefore in agreement with the high delay detected by the PCMCI method for this effect. The effect may also be explained by radiative warming of exposed ocean surface (Sévellec et al., 2017; Jenkins and Dai, 2022).

GCSS is not suited to be applied to this example, because it does not offer the option of systematically masking the data, and thus, wanting to understand seasonally constrained interactions, cannot be applied here. LKIF lacks several causal effects, as it struggles to detect delayed effects and implements masking in a way that drastically reduces available sample size. With PCMCI, a stable causal network is detected. The strength estimation of effects derived from it, however, likely underestimates the magnitude of the interactions, mainly because much of them may happen on slower time scales, as suggested by model studies (Mahajan et al., 2011; Li et al., 2021; Liu and Fedorov, 2022), which we are not able to analyze due to limited data availability on longer time scales. The short delays on monthly time scales assessed here, can only capture relatively fast parts of the (assumed) physical interactions, leaving more delayed and more continuous interactions (e.g., the meltwater fluxes from different regions of the Arctic Ocean arriving in different months), as well as long-term responses out of the picture.

Furthermore, the impact of an AMOC weakening on oceanic heat transport appears to be nonlinear, i.e., while the current cold anomaly in the North Atlantic is largely constrained to the subpolar gyre region (Caesar et al., 2018), projections of an



AMOC shutdown predict sea surface temperatures to drop massively across the Arctic and subpolar seas (van Westen et al., 2024), which may well have a larger effect on Arctic sea ice than a linear extrapolation of current observations would imply.

In future analyses, the inclusion of larger time scales could add to the already observed effect strength, however, this would require using data from ESMs, requiring experiments that extend past 2100. Model data is generally an interesting future field to apply causal analysis to tipping processes and interactions.

## 6 Conclusions

All three causal inference methods analyzed in this paper can be reliable tools for the detection of interactions of climate
tipping elements. However, we determined several conditions for sufficient reliability that need to be considered for their application. Researchers should take adequate care that sufficient sample sizes are available, while sampling on a time scale roughly matching expected interaction delays. Strong nonlinearities in data, e.g., from tipping processes, reduce the reliability of the methods significantly. The choice of a causal method is crucial and depends to a large degree on known conditions: The GCSS method appears most useful in large datasets, or if significant delay effects can be expected. The LKIF method is most
reliable for application cases on the lower end of data availability and interaction strengths, while the PCMCI method shows weaker detection power across most parameterizations, but offers enhanced flexibility for the inclusion of expert knowledge.

Although this work covers several common challenges in real data, the behavior of physical systems remains more complex than the idealized models we analyzed. Such phenomena include processes on multiple time scales of systems and their interactions, or seasonality and regime dependence in data.

Further research could identify approaches and recommendations on causal analysis in the presence of tipping events or how expert knowledge can be integrated into causal analysis to tackle the additional complexity of physical systems.

In an applied experiment, we utilized the PCMCI method to detect a bidirectional stabilizing interaction between Arctic summer sea ice and the AMOC on reanalysis data. Despite constraints in terms of data availability, we find that these interactions resemble the physical effects suggested by domain experts and model experiments. Due to a lack of long-term observational
data on Arctic sea ice, we consider it the most promising approach for further investigations to apply the presented causal methods to data from ESMs to explore the role of time scales and regime shifts under global warming on these interactions.

Overall, these results encourage the usage of causal inference methods for climate tipping elements under careful consideration of the data availability and assumptions on the observed processes.

*Code and data availability.* All data sets used for the applied experiment are publicly available. The AMOC fingerprint is based on ERA5
data, see Hersbach et al. (2023), Arctic sea ice data is taken from E.U. Copernicus Marine Service Information (CMEMS). Marine Data Store (MDS) (2024b), and Arctic temperatures from E.U. Copernicus Marine Service Information (CMEMS). Marine Data Store (MDS) (2024a). Code for the experiments on synthetic data and for causal analysis (Lohmann, 2025) is available at the DOI: 10.5281/zenodo.17864597. We use the PCMCI+ algorithm from the *Tigramite* package by (Runge et al., 2019b), for the LKIF algorithm we use the package *LK_Info_flow*



by Rong (2024) following an implementation by Liang (2021). For the GCSS algorithm, we use the original Matlab code by the authors
(Barnett, 2020), for which we implemented a significance test manually.





## Appendix A: Interaction Networks

Figure A1 shows six network systems that were manually designed with a focus on stability in the absence of forcing ($c = 0$ in Eq. 1). For numbers of variables that are not covered by these systems, networks of lower sizes are derived by removing nodes from the next larger network, starting with the node with the highest index.

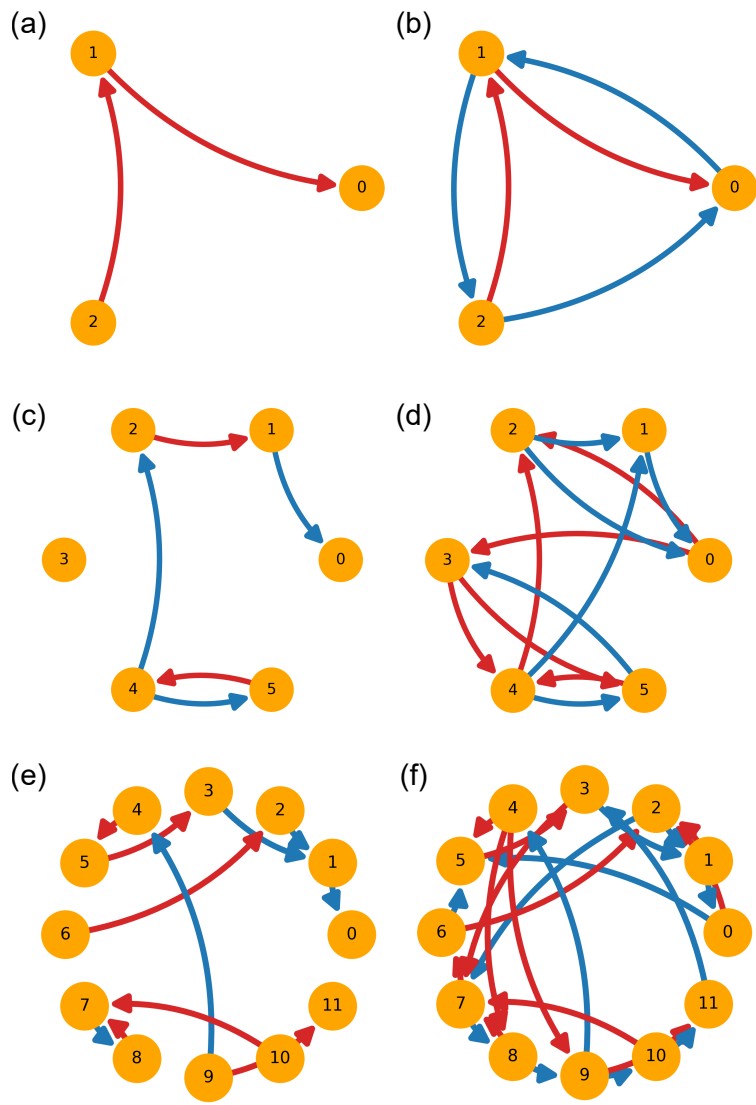

**Figure A1.** The different systems used for synthetic data generation, with varying numbers of variables and two density levels (where (a,c,e) are sparse systems, and (b,d,e) are dense). Red edges indicate positive interaction strength, blue edges indicate negative interaction strength.



## Appendix B: Fitting Models to Data

Both the LKIF and the GCSS method are based on explicitly fitting a model to the input data, each of which assumes linearity in some form. As we use PCMCI with the partial correlation test, the method implicitly assumes a linear impact of interactions on variables, too.

Therefore none of the models assumed by these methods fit exactly to the synthetic data generation model with a cubic differential equation. However, two additional assumptions about the data generation process can fix this issue.

- Firstly, if the system is stationary, i.e., no forcing is applied and the system is in equilibrium from the start, then the system can be approximated by a linearization around the equilibrium (i.e., a first-order Taylor expansion). Equation 1 in its linearized form around $x = 1$ with $c = 0$ thus becomes:

$$\dot{\Delta x_i} = -2\Delta x_i + \sum_{j \neq i} s_{j,i} \Delta x_j + \sigma dW, \tag{B1}$$

where $\Delta x$ describes the deviation of a variable from its equilibrium (here in $x = 1$). Note that this model aligns with the model employed by the LKIF algorithm. However, large noise may move the state out of the proximity of the equilibrium that is approximated well. Additionally, the interactions between variables technically violate the stationarity assumption (i.e., noise in one variable can move the equilibrium of another variable around which its linearization was conducted).

- Secondly, time discretization is required to numerically integrate the model. The GCSS method further explicitly assumes a time-discrete process for the underlying data. Any linear ordinary differential equation can be approximated by a time-discrete vector autoregressive process. We denote a derivation with the Euler forward integration method without stochastic terms for simplicity.

$$\Delta x_{i,t+\Delta t} = \Delta x_{i,t} + \Delta t(-2\Delta x_{1,t} + \sum_{j \neq i} s_{j,i} \Delta x_{j,t}). \tag{B2}$$

With a suitable choice of $\Delta t$, this procedure can approximate the original process with arbitrary accuracy (again, given there is no stochastic element). This linear, time-discrete model transformation aligns with the model of GCSS, and fulfills the model assumptions of PCMCI. Since there is an inherent delay effect between the values of a causal parent and a causal child in a linear differential equation, we choose $\Delta t = 0.1$ for our experiments as a tradeoff between accuracy and this implicit delay length.





## Appendix C: Parameterization of Synthetic Data Generation

Table C1 lists the parameters of the synthetic data generation and of the causal inference methods (error rate and max. time lag) in their full ranges for the corresponding experiments and their default values where they are not varied explicitly.

**Table C1.** Parameters used in the experiments with their corresponding ranges

| Parameter name | Full test values | Default value |
|---|---|---|
| Delay length | 0, 0.1, 0.2, 0.3, 0.4, 0.5, 0.6, 0.7, 0.8, 0.9, 1.0, 1.5, 2.0, 3.0 | 0 |
| Number of samples | 50, 100, 200, 500, 1000, 2000, 5000, 10000 | 1000 |
| Coupling strength | 0.1, 0.2, 0.5, 0.7, 1.0, 1.5, 2.0, 2.5, 3.0 | 1.0 |
| Forcing strength | 0, 0.1, 0.2, 0.3, 0.4, 0.5, 0.6, 0.7, 0.8, 0.9, 1.0 | 0 |
| Noise scale | Fixed | 0.01 |
| Error rate $\alpha$ | Fixed | 0.05 |
| Max. time lag | Fixed | Delay length + 1 |





## Appendix D: Conditional Independence Tests in PCMCI

As the PCMCI method provides several different conditional independence tests, Fig. D1 displays the results of the sampling experiment (from Fig. 2a) with the robust partial correlation test, which normalizes the distribution of samples to fulfill the assumptions of partial correlation tests more closely, and with the nonlinear Gaussian process distance correlation (GPDC) test Runge et al. (2019b). As the GPDC test is significantly more computationally intensive, we only conduct this experiment with up to 2000 samples.

The differences in detection scores between the analyzed conditional independence tests is very small and lies within half a standard deviation, with the partial correlation test yielding a slightly higher MCC.

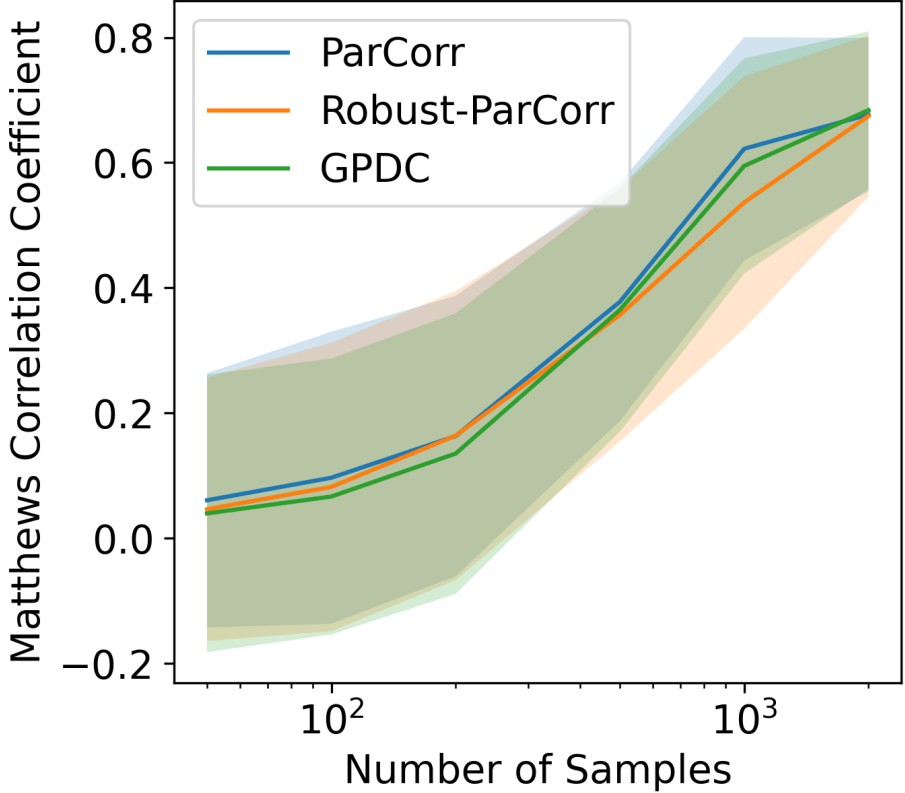

**Figure D1.** The performance of PCMCI with the three compared conditional independence tests is highly similar, with a small advantage of the partial correlation test over the other two.





**Appendix E: Runtimes of Causal Methods**

When applied to a small number of variables as in this study, all presented causal algorithms can be run on standalone hardware for singular executions. Here we present statistics of their runtime over 100 runs at the default settings of Tab. C1 on the PIK high performance computer system, using AMD EPYC 9554 Genoa processors at 3.1GHz.

The GCSS algorithm performs with a very consistent runtime of about $10^{-2}$ seconds, while the LKIF method shows large
outliers with higher runtimes, but a median runtime even lower than that of GCSS. PCMCI (with the partial correlation test) is the slowest algorithm and takes more than $0.1$ seconds per run.

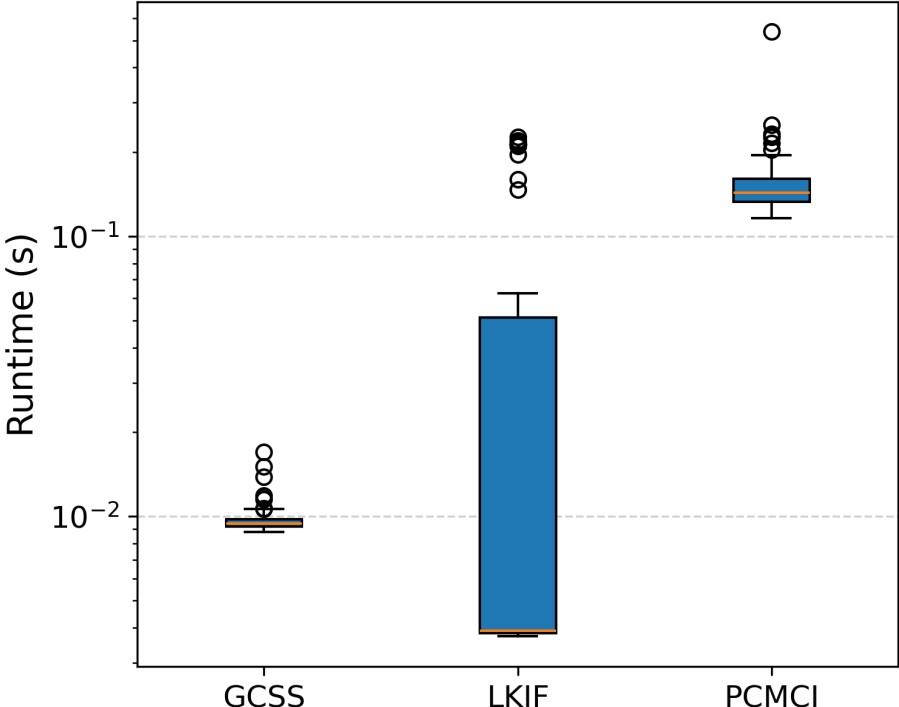

**Figure E1.** The runtime statistics of the presented causal methods. The orange line indicates the median, the blue bar mark values between the 25th and 75th percentile, and whiskers extend to the outermost data point that is within $1.5$ times the interquartile range (i.e., the size of the blue bar). Outliers beyond the whiskers are represented by circles. While LKIF is the fastest method by its median, it shows large deviations up to one order of magnitude. The median runtime of GCSS is less than an order of magnitude larger than that of LKIF, and GCSS is more consistent, while PCMCI is significantly slower than both.




**Appendix F: Extended Experiments on Arctic Sea Ice and AMOC**

We conduct further experiments to test the robustness of the detected interactions between Arctic summer sea ice and the AMOC. Firstly, the LKIF method is applied, and confirms only the link from the AMOC to Arctic summer sea ice, see Fig.

F1a. Secondly, several alternative data sources for sea surface temperatures are used to determine the AMOC fingerprint as established by Caesar et al. (2018): Data from HadCRUT5 (Morice et al., 2021), HadISST4 (Kennedy et al., 2019), COBE-SST2 (Hirahara et al., 2014) and ERSSTv5 (Huang et al., 2017) are aggregated and preprocessed in the same manner as the ERA5 data in the main part of this study. These datasets were prepared by (Högner et al., 2025) and are used here unchanged. The causal analysis shows highly similar results for all but the HadCRUT5 data set, see Fig. F1b-e. While the slower and

weaker destabilizing link from Arctic summer sea ice to the AMOC is not confirmed by these robustness tests, they provide strong evidence that the main mechanisms identified in our experiment are robust to observational noise and differences in reanalysis procedures.

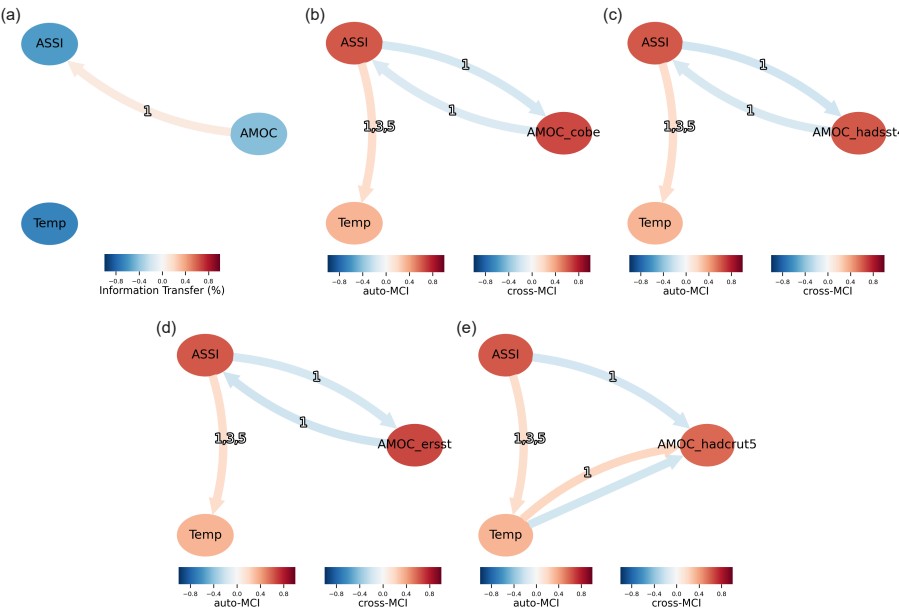

**Figure F1.** The causal networks as determined by (a) the LKIF method shows a single causal link from Arctic temperatures to the AMOC. (b-d) With several data sources of sea surface temperatures other than the ERA5 dataset used in the main part of this study, the results are mostly robust, only the slow destabilizing link from Arctic summer sea ice to the AMOC is not confirmed with any of the four additional datasets. (e) The AMOC fingerprint based on HadCRUT5 data shows an effect of Arctic temperatures on the AMOC, but appears as an outlier in these robustness tests.



*Author contributions.* N.L., D.S., M.B. and N.W. conceptualized the study; N.L. and N.W. designed the study; N.L. conducted implementation and analysis; A.H. provided data for the applied experiment; N.L. designed the figures with input from all authors, N.L. led the writing
with input from all authors; N.W. supervised the study.

*Competing interests.* The authors declare that they have no conflict of interest.

*Acknowledgements.* N.L. and N.W. are grateful for funding from the Klaus Tschira Foundation (under the grant agreement ID 25545). The authors gratefully acknowledge the European Regional Development Fund (ERDF), the BMFTR and the Land Brandenburg for supporting this project by providing resources on the high-performance computer system at the Potsdam Institute for Climate Impact Research. D.S.
acknowledges support from the Alexander von Humboldt Foundation in the framework of the Alexander von Humboldt Professorship endowed by the German Federal Ministry of Education and Research (BMBF).





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
