# Peer review of "Quantitative Comparison of Causal Inference Methods for Climate Tipping Points"

_EGUsphere, 2025_

## Referee Comment (RC1)

**Review: Quantitative Comparison of Causal Inference Methods for Climate Tipping Points**

**General Comments:**

In this work, the authors conduct a quantitative investigation into the reliability and robustness of three multivariate causal inference methods:

1. Liang–Kleeman Information Flow (LKIF)
2. Peter–Clark Momentary Conditional Independence (PCMCI)
3. Granger Causality for State Space Models (GCSS)

This is done in the context of studying the interactions of climate tipping elements in various facets of the Earth system which pose specific operational challenges. Through the quantitative metric of choice (Matthews Correlation Coefficient; MCC), the authors showcase unique advantages for each method, while also identifying three general principles for addressing nonlinear responses, delayed effects, and confounders during the application of these causal methods to climate tipping points. The use of MCC is natural and justified, as it considers balanced ratios of the confusion matrix in binary classification.

Following a preliminary study on synthetic data generated by a network of differential equations, they apply LKIF and PCMCI, based on their recommendations, on reanalysis data to detect tipping point interactions between Atlantic Meridional Overturning Circulation (AMOC) and Arctic summer sea ice (ASSI), confirming established physical mechanisms (bidirectional stabilizing interactions) beyond confounding influences (Arctic temperatures).

This study is a welcome addition to both the climate tipping literature and to the causal inference community. The structure of the paper is sound, transitioning from a synthetic-data investigation to a realistic application to climate tipping point interactions between AMOC and ASSI. Physically consistent results are derived in the latter study, both in terms of state-space causality and temporal causal influence regions, by applying the recommendations derived from the former experiment.

My general assessment is that this is a well-written paper overall, with the authors presenting their methodology and results succinctly and clearly, which should be of interest to the relevant researchers. The results are put in context, well interpreted, and presented without drawing strong conclusions. This work fits into the scientific scope of NPG. My recommendation is that it can be published to NPG following some major revisions and clarifications, as well as some minor corrections and adjustments.

**Specific Comments:**

(Format: p.##, l.## - Page number, line number | Section/Appendix/Figure/Table ##)

p.2, l.32—44 – In terms of references, the authors appropriately cite most relevant works in the associated fields throughout the manuscript. But, while the authors succinctly explain climate tipping points and provide constructive and relevant examples here, I would recommend that they note, either implicitly or explicitly, how they essentially describe bifurcation-generated tippings here (with a hint towards rate-induced tips when referring to effects across time scales), with other regime-switching driving mechanisms (internal variability and rate-limited tipping) also being possible **[1]**.

**[1]** https://arxiv.org/abs/1103.0169

p.7, l.178—179 – I would recommend elaborating a bit more on the details of how the time lag analysis is calculated in LKIF ("Time lag analysis is implemented by shifting any single input time series by a given number of time steps.") and whether the adopted approach is operationally consistent with the approaches in PCMCI and GCSS. These details can be added in the appendix if preferred.

Section 2.3 – I would recommend adding a very brief paragraph here providing an interpretation of MCC and its values for people not familiar with the metric (e.g., maximum values and zero values, relation to the chi-squared statistic or other scores for intuition, etc.), which would also help with the self-containedness of the work.

p. 9, l. 223—227 (and p.21, l.481—483 by extension) – As a quick clarification, how are the results affected by a different heuristic choice of the time step $\Delta t$ to account for causal delays? Specifically, how are the results in Panel (b), Figure 2 affected by implicitly changing the signal-to-noise ratio of each relation? A quick note here would also help with empirically elaborating on Recommendation 1 in Section 4.

p.10, l. 239—240 – Indeed, since GCSS assesses causal relationships by projecting the latent state process (cause) onto the space spanned by the infinite past of the observation variables with and without the effect, it provides higher explanatory power with the autoregressive part resolving the presence of time lags. Elaborating a bit more on this argument will make the justification slightly more rigorous.

p. 14, l. 325 – A quick note on why GCSS cannot be implemented here in a straightforward manner due to implementation details would be welcome (the small number of samples available for this experiment is also a valid limitation, based on the results of Fig. 2a). I do note the additional clarification in p. 17, l. 412—413, which is more than enough, but it does come towards the end of the case study.

p.14, l. 341—343 – Is the choice of including cells above the 66$^{th}$ percentile based on a specific heuristic in the associated literature? How sensitive are the results to this choice?

Figure 5 and p.15—16, l. 369—389 – I would recommend a clarification of the results in Figure 5 and the associated text: The colour of the causal arrow from ASSI to AMOC indicates a stabilizing/negative effect at one month delay but at five months delay there's a destabilizing/positive link instead. While that is the weaker link, as the text notes, adding two arrows that are independently coloured instead of single one that is coloured with respect to the stronger link would remove any ambiguity or confusion. I would recommend the same for the causal effect from ASSI to the Arctic temperatures (adding three arrows). That way, the coefficient of each link can also be superimposed next to the corresponding arrow for clarity. If the authors choose to implement these changes, they can also apply them to Figure F1 for consistency.

Table C1 – For the synthetic data experiments, has a larger (or non-uniform) noise level been tested (but not too large as to break the required assumption of stationarity through the linear couplings)? Also, I would recommend adding the variable next to the parameter name in the first column, which would also clarify the use of uniform coupling strengths (not considering the choice of sign). E.g., "Noise scale ($\sigma$)".

Appendices D, E, and F – Just wanted to note that these are a very nice addition to the text, further illuminating the implementation intricacies behind the causal methods utilized in terms of different variants, operational complexity, and robustness to observational noise and different reanalysis approaches, respectively.

**Technical Corrections:**

(Format: p.#, l.# - Page number, line number)

1. p.1, l.9 – The LKIF abbreviation is used in l.14 of the abstract but not defined here.
2. p.3, l.50—53 – If possible, the authors can slightly revise this sentence for clarity and readability.
3. p.3, l.59 – The PCMCI acronym is used here but first defined in l.77.
4. p.4, l.104 – Small typo: "...not known to the causal method **and** introduces common...".
5. p.5, l.116 – As a small note, since the Wiener processes for each state variable are mutually independent (as also noted in the text), please consider adding an appropriate subscript to $W$ to indicate this (I assume the diffusion feedback $\sigma$ is held constant across $x_i$).

6. p.5, l.117 – As a minor note, I would recommend first noting here the role of $c$ as the common confounder in the induced causal diagrams, akin to p.10, l.255—262, which would also clarify Panel (a) in Figure 1.

7. p.5, l.121—123 – It could be that I'm missing something, but shouldn't the cited negative critical value correspond to the transition threshold in the absence of the additive noise and linear coupling terms? If yes, maybe consider slightly rephrasing this sentence to avoid ambiguity.

8. p.6, l. 140 – Small typo: "…are established in **the** literature.".

9. p.7, l. 163 – Small typo: "…can be found in **Appendix** D.".

10. p.10, l. 239 – Small typo: "…for time lags**. W**e consider…".

11. p. 10, l. 258 – For clarification: "…(without interactions **and noise**).".

12. p. 10, l. 260—279 – "exclusion" and "inclusion" are used here for the confounder term, but "hidden" and "known" are used in the legend of Figure 4. I would recommend sticking to the former throughout for consistency.

13. p. 10, l. 263—266 – I could be misinterpreting Figure 4, but I think this excerpt should read as "…**for the LKIF algorithm in the absence of forcing**.", "…**the forcing strength does not have an influence on the true positive rate of LKIF**.", and "…GCSS drops for an unknown confounder, but **the false positive rate** remains unchanged if the confounder is included in the causal analysis.".

14. p. 12, Figure 4 Caption – For clarification: "The LKIF method **instead** sees a large decline in false positives **and** the PCMCI method does not show a clear effect".

15. p. 13, Section 4 – Referencing the relevant panels from Figures 2—4 here would help for fast lookup.

16. p.13, l. 293 – Small typo: "…as described in **Appendix** B.".

17. p.15, l. 346 – Small typo: "…(Carvalho and Wang, 2020)**. A** reduction…".

18. Panels (e) & (f), Figure A1 – Some connections (e.g., 9→11) might be construed as being moderated by an intermediate variable (10 in this case). While this shouldn't be an issue considering the structure of the linear and explicit couplings in Eqs. (1) and (2), having the arrows circumvent the extraneous nodes in the diagram would leave no room for misinterpretation. Finally, I would recommend noting Panel (c) as the default model network for clarity, just like the last column of Table C1.

19. p.21, l.470 – I would recommend writing "$\Delta x_i = x_i - 1$" here for simplicity.

20. p.21, l.478—483 – Small typo: $\Delta x_{1,t}$ should read as $\Delta x_{i,t}$ in Eq. (B2). I would also recommend including the noise term (using Euler—Maruyama), making (B2) a coupled VAR(1) process that is consistent with the preceding exposition, as the simplification without the stochastic term does not really simplify things that much. The approximation statement in l.479—480 is still true then, both weakly and strongly (under appropriate convergence orders).

---

## Referee Comment (RC2)

[referee-annotated manuscript omitted]